# Hybridization between *Crotalus aquilus* and *Crotalus polystictus* Species: A Comparison of Their Venom Toxicity and Enzymatic Activities

**DOI:** 10.3390/biology11050661

**Published:** 2022-04-26

**Authors:** Octavio Roldán-Padrón, Martha Sandra Cruz-Pérez, José Luis Castro-Guillén, José Alejandro García-Arredondo, Elizabeth Mendiola-Olaya, Carlos Saldaña-Gutiérrez, Patricia Herrera-Paniagua, Alejandro Blanco-Labra, Teresa García-Gasca

**Affiliations:** 1Laboratorio de Biología Celular y Molecuar, Facultad de Ciencias Naturales, Universidad Autónoma de Querétaro, Av. de las Ciencias S/N, Juriquilla, Queretaro 76230, Qro, Mexico; octavio_rolpad@hotmail.com (O.R.-P.); martha.sandra.cruz@uaq.mx (M.S.C.-P.); carlos.saldana@uaq.mx (C.S.-G.); patricia.herrera@uaq.mx (P.H.-P.); 2Departamento de Biotecnología y Bioquímica, Centro de Investigación y Estudios Avanzados del IPN, Km. 9.6 Libramiento Norte Carr. Irapuato-León, Irapuato 36824, Gto, Mexico; tekk.el.527@gmail.com (J.L.C.-G.); elizabeth.mendiola@gmail.com (E.M.-O.); 3Laboratorio de Química Medicinal, Facultad de Química, Universidad Autónoma de Querétaro, Cerro de las Campanas S/N, Centro Universitario, Queretaro 76010, Qro, Mexico; alejandro.gr@uaq.mx

**Keywords:** rattlesnake, venom enzymes, proteases, hyaluronidases, phospholipase A_2_, hybridization

## Abstract

**Simple Summary:**

When two different species or subspecies of animals have progeny, we speak about hybrid organisms, which present a mixture of the genetic characteristics of their parents. This phenomenon occurs in nature in a common way, although most of the time the hybrids between species are sterile. In this work, the venom characteristics of hybrids from two species of rattlesnakes were studied: *Crotalus aquilus* (father) and *Crotalus polystictus* (mother), both endemic to central Mexico. Scale numbers (phenetic analysis) and venom protein were compared between hybrids (females and males), biological parents, and adult individuals of the two species. The presence and activity of the main types of enzymes in these venoms were analyzed, and the lethal dose was determined in mice. Through the phenetic analysis, it was observed that the hybrids were more similar to *C. polystictus* (mother), the presence of proteins and enzymatic activity resulted in a combination of the two species, but the lethality of the venom was greater in the hybrids. These results allow us to learn more about the way in which the hybridization phenomenon influences the characteristics of rattlesnake venom. Some of the applications of this knowledge could be used to develop more effective antidotes.

**Abstract:**

Hybridization is defined as the interbreeding of individuals from two populations distinguishable by one or more heritable characteristics. Snake hybridization represents an interesting opportunity to analyze variability and how genetics affect the venom components between parents and hybrids. Snake venoms exhibit a high degree of variability related to biological and biogeographical factors. The aim of this work is to analyze the protein patterns and enzymatic activity of some of the main hemotoxic enzymes in snake venoms, such as serine proteases (trypsin-like, chymotrypsin-like, and elastase-like), metalloproteases, hyaluronidases, and phospholipase A_2_. The lethal dose of 50 (LD_50_) of venom from the *Crotalus aquilus* (Cabf) and *Crotalus polystictus* (Cpbm) parents and their hybrids in captivity was determined, and phenetic analysis is also conducted, which showed a high similarity between the hybrids and *C. polystictus*. The protein banding patterns and enzymatic activity analyze by zymography resulted in a combination of proteins from the parental venoms in the hybrids, with variability among them. In some cases, the enzymatic activity is higher in the hybrids with a lower LD_50_ than in the parents, indicating higher toxicity. These data show the variability among snake venoms and suggest that hybridization is an important factor in changes in protein concentration, peptide variability, and enzymatic activity that affect toxicity and lethality.

## 1. Introduction

Snake venoms are a mixture of proteins, peptides, and toxins that vary extensively between and within snake species [1]. This variability is widely reported at different taxonomic levels, from the family level down to variations between organisms of the same species. They are also related to geographical location or type of feeding and to other variables, such as age-dependent changes, sexual dimorphism, and factors associated with feeding type, which can affect the degree of the venom’s lethality; however, the mechanisms that cause variation in venoms remain mostly unknown [2,3]. Nevertheless, the processes of gene duplication and positive selection appear to be the predominant mechanisms for generating diversity in snake venoms [1]. The variation in the composition of venoms is of particular interest in terms of their mechanisms of action [4,5], as well as their medical implications [6] and consequences for the efficacy of snakebite treatments [1]. Another phenomenon associated with gene variation is hybridism [7,8], which can be defined as the interbreeding of individuals from two populations, or groups of populations, that are distinguishable by one or more heritable characteristics, or the incorporation of genes from differentiated populations into another. This phenomenon has been reported in several organisms as an adaptive variable [8,9], usually associated with geographical areas where populations of nearby species are found [10] and could be related to the mechanism that contributes to speciation [11,12,13]. In the case of snakes, hybridism is also associated with variability in venom composition [2,3]. Hybridization in snakes has been reported based on analyses of morphological and genotype features [11]. In rattlesnakes, hybridization was first documented by Bailey [14], who identified a hybrid case between *Crotalus horridus* and *Sistrurus catenatus* by conducting a thorough examination of scalation, body proportions, and color patterns. There are just a few works reporting the biological and enzymatic activities of the venoms in hybrids from *C. atrox* and *C. scutulatus* [15], *Bothrops erythromelas* and *B. neuwiedi* [11], *C. s. scutulatus* and *C. oreganus helleri* [16], *C. viridis* and *C. s. scutulatus* [3], and *Protobothrops flavoviridis* and *P. elegans* [17].

During 2006 and 2007, seven hybrid organisms were born from two hybridization events that occurred between a captive individual of *Crotalus aquilus*, the biological father (Cabf), and *Crotalus polystictus*, the biological mother (Cpbm), at the Animal Resource Facility (Autonomus University of Querétaro, México). Both species are endemic to central Mexico [18,19,20]. These rattlesnakes are part of the Viperidae family [21,22], with hemotoxic venoms that act directly on the cells and hemostatic system, causing extensive local lesions, hemorrhages, edema, tissue myonecrosis, and a variety of alterations in blood coagulation [23,24]. Such damage is the result of the activity of several components of the venom, including metalloproteases, serine proteases, hyaluronidases, phospholipase A_2_, L-amino oxidases, and acetylcholinesterase, as well as non-enzymatic proteins such as type C lectin proteins and other minor components [25,26]. 

In this work, the main hemotoxic enzymatic activity types (serine proteases, metalloproteases, hyaluronidases, and phospholipase A_2_) were compared, as well as the LD_50_ in mice, and the differences between venom patterns were obtained by reverse-phase HPLC, MALDI-TOF-MS. On the contrary, a phenetic analysis, based on the number and morphological characteristics of the scales, was also determined to establish the similarity between the parental species and hybrids.

## 2. Materials and Methods

### 2.1. Phenetic Analysis

Captive male *Crotalus aquilus* and female *Crotalus polystictus* rattlesnakes and their hybrids born in 2006 and 2007 were housed and cared for in the Animal Resource Facility (Autonomous University of Querétaro, Querétaro, México). Phenetic analysis was performed comparing the number of dorsal spotting and tail band patterns, and the characteristics of the number of scales around the rattle, subcaudal, ventral, supralabial, infralabial, dorsal, and intersupraocular were also counted from the following juvenile hybrids (250–325 mm long) of the first hybridization (four years old) in 2006: Male hybrid 1 (MH1); female hybrid 2 (FH2); female hybrid 3 (FH3); male hybrid 4 (MH4); juvenile hybrids of the second hybridization (three years old) in 2007: Female hybrid 5 (FH5), male hybrid 6 (MH6), and male hybrid 7 (MH7); three males of *C. aquilus*: Two juveniles, MCa8 and MCa9 (250–325 mm long), and one adult (>450 mm long) that was the biological father (Cabf); and two *C. polystictus* adults (>450 mm long), a male (MCp10) and the biological mother (Cpbm). A second analysis was performed using the characteristic of the number of scales plus the characteristic of the protein bands, using the two protein bands of Cabf (26 kDa) and Cpbm (31 kDa) from the SDS-PAGE analysis. Each characteristic was taken as an operational taxonomic unit (OTU). These data were used to form a similarity matrix, analyzed by the taxonomic similarity coefficient to clustering similar groups using the unweighted pair group method with the arithmetic mean (UPGMA) method in the NTSYSpc program version 2.2 [27].

### 2.2. Venom Harvest

The venom used for the determination of serine proteases (trypsin- and chymotrypsin-like), metalloproteases (zymography), and the LD_50_ was obtained by manual extraction (milking) of 11 rattlesnakes, seven of which were from the four-year-old hybrids of the first hybridization (MH1, FH2, FH3, and MH4) and the three-year-old hybrids; the following three of the second hybridization event (three years old): FH5, MH6, and MH7; the following two male adults of *Crotalus aquilus*: MCa8 and the biological father Cabf; and two adults of *C. polystictus*, MCp10 and the biological mother Cpbm. The venom was immediately stored on ice, lyophilized, and then kept at −70 °C until use, only during the first three days after milking. In 2017, Cabf, Cpbm, and two hybrids (FH2 and FH5) died, making it necessary to determine the enzymatic activity of elastases, metalloproteases, phospholipase A_2_, and hyaluronidases from the venom obtained from five adult hybrids, namely, three 11-year-old hybrids (MH1, FH3, and MH4) and two 10-year-old hybrids (MH6 and MH7), as well as the pooled venom of four male adults of *Crotalus aquilus*, MCa-PV, and pooled venom of four female adults of *Crotalus polystictus*, FCp-PV.

### 2.3. Polyacrylamide Gel Electrophoresis 

The lyophilized venom was diluted in tridistillated water and the protein concentration was determined [28] using as protein standard BSA (Sigma-Aldrich, catalog number 05470, St. Louis, MO, USA) at different concentrations. SDS-PAGE using 20 µg of venom protein was analyzed under reducing (4% of β-mercaptoethanol and 5 min at boiling water temperature) and non-reducing conditions [29], using a 10% or 12% polyacrylamide gel in a Mini Protean II unit (Bio-Rad Hercules, CA, USA) or a Hoefer *MiniVE system* (Hoefer, Holliston, MA, USA). The protein bands were stained with Coomassie Blue G250 dye (Bio-Rad, catalog number 1610406, CA, USA), unstained with a 40% and 10% methanol–acetic acid solution, respectively, and imaged on an HP Scanjet 4570c scanner.

### 2.4. Zymography Assays

#### 2.4.1. Serine Proteases Zymography

Proteolytic activity measured by zymography was detected based on the methodology reported by Ohlsson et al. [30] and Vinokurov et al. [31]. For trypsin and chymotrypsin zymography, after electrophoresis of 50 and 100 µg of venom protein, respectively, the gel was washed with 0.1 M Tris-HCl pH 8 buffer and then a cellulose membrane (Bio-Rad, catalog number 1620112, CA, USA) previously embedded in the corresponding substrate, BA*p*NA (Sigma-Aldrich, catalog number B4875, St. Louis, MO, USA) or SAAPF*p*NA (Sigma-Aldrich, catalog number S7388, St. Louis, MO, USA), respectively. Both substrates were previously prepared in dimethylformamide and diluted 1:20 (*v*/*v*) in 0.1 M Tris-HCl to obtain a final concentration of 1 mM. The membranes were placed on top of the electrophoresis gel and incubated at 37 °C for 2–2.5 h or until bands with yellow coloration appeared. The cellulose membrane was serially washed for 5 min with sodium nitrite (NaNO_2_) 0.1% in HCl 1 M, ammonium sulfate (NH_4_SO_3_NH_2_) 0.5% in HCl 1 M, and 0.05% *N*-(1-naftil) ethylenediamine dihydrochloride (C_12_H_14_N_2_ 2HCl) in 47.5% ethanol. After incubation, the gels were stained with Coomassie Blue G250 dye and unstained with 10% acetic acid. 

#### 2.4.2. Metalloprotease Zymography

An analysis of 20 µg of the venom protein samples was conducted in a 10% or 12% SDS-PAGE gel co-polymerized with 0.12% gelatin (Sigma-Aldrich, catalog number G2500, St. Louis, MO, USA) as the substrate [32]. After electrophoresis, the gel was washed in a 2.5% (*v*/*v*) Triton X100 solution for 1 h and later for 30 min in a 0.05 M Tris-HCl pH 7.4 buffer, then finally incubated for 2–3 h in a 0.2 M NaCl, 0.005 M CaCl_2_, 0.002% (*v*/*v*) Triton X-100, 0.001 M cysteine, and 0.05 M Tris-HCl pH 8 buffer. After incubation, the gel was stained with Coomassie Blue G250 dye and then unstained with 40% methanol–10% acetic acid. 

#### 2.4.3. PLA_2_ Zymography

Twenty micrograms of the venom protein samples were run in 10% SDS-PAGE gel under non-reducing conditions; then, the gel was washed for 1 h with 2.0% (*v*/*v*) Triton X-100 and 0.5 M Tris-HCl pH 7.4 buffer. The gel was subsequently washed for 30 min in a buffer with 140 mM NaCl, 2.5 mM CaCl_2_, and 50 mM Tris-HCl pH 7.4 buffer and then incubated for 2.5 h at 37 °C on a 1% agarose (Sigma-Aldrich, catalog number A9539, St. Louis, MO, USA) gel co-polymerized with a solution containing 50 mM Tris-HCl (pH 7.4), 140 mM NaCl, 2.5 mM CaCl_2_, and 2% egg yolk. Light zones indicate the presence of PLA_2_ [33].

#### 2.4.4. Hyaluronidases Zymography

A 10% SDS-PAGE gel was co-polymerized with 1 mg/mL of hyaluronic acid (Sigma-Aldric, catalog number H7630, St. Louis, MO, USA), and then 100 µg of the venom protein samples was analyzed via non-reducing electrophoresis. The gel was washed for 30 min with 0.2 M sodium acetate (pH 6), 0.15 M NaCl, and 2.5% (*v*/*v*) X-100 Triton buffer solution and subsequently incubated for 2.5–3 h at 37 °C in the same buffer solution without Triton. The gel was stained with alcian blue dye (Sigma-Aldrich, catalog number A5268, St. Louis, MO, USA) at 0.5% (*w*/*v*) and subsequently de-stained with a solution containing 40% methanol and 10% acetic acid [34].

### 2.5. Enzymatic Assays

#### 2.5.1. Serine Proteases

The proteolytic activity was determined according to the method reported by Erlanger et al. [35]. Briefly, 20 μL of Trypsin and chymotrypsin-like substrates BA*p*NA or SAAPF*p*NA were used at 0.01 M in dimethyl sulfoxide (DMSO), and then 20 μL of the venom sample was added and carried to a final volume of 240 μL in 0.1 M Tris-HCl buffer pH 8. The reaction was incubated for 2 h at 37 °C. For the elastase-like activity, 10 μL of N-methoxysuccinyl-Ala-Ala-Pro-Val *p*-nitroanilide (Sigma-Aldrich, catalog number M4765, St. Louis, MO, USA) at 0.01 M in DMSO was used and carried to a final volume of 120 μL in 0.1 M Tris-HCl buffer pH 5. Trypsin-like activity was determined using 20 μL of venom sample from a mix of 1 μL of crude venom diluted in 99 μL of distilled water. Chymotrypsin-like activity was determined using 20 μL of the venom sample from a mix of 1 μL of crude venom diluted in 999 μL of distilled water. For elastase-like activity, 50 μg of the protein venom sample was used. The enzymatic activity was expressed as specific activity, calculating the amount of protein in each sample according to the Bradford method. The absorbance of liberated *p*-nitro aniline of the mixture was measured in a spectrophotometer (Benchmark Plus, Bio-Rad, USA) at 405 nm and 37 °C in a 96-well plate. A unit of proteolytic activity is defined as an increase of 0.01 absorbance at 405 nm per min, while specific peptidase activity is expressed as units of proteolytic activity per microgram of protein venom. The experiments were carried out in triplicate. In each case, bovine trypsin (Sigma-Aldrich, catalog number T1426, St. Louis, MO, USA), bovine chymotrypsin (Sigma-Aldrich, catalog number C4129, St. Louis, MO, USA), and porcine elastase (Sigma-Aldrich, catalog number E1250, St. Louis, MO, USA) were used as positive controls. 

#### 2.5.2. Metalloprotease Proteolytic Activity

Proteolytic activity was assayed according to Gutiérrez et al. [36] and Ponce et al. [37] with modifications. Briefly, total enzymatic activity was measured in the first set of tubes where 100 µg of venom protein and 500 µL of 2% (*w*/*v*) casein (Sigma-Aldrich, catalog number C3400, St. Louis, MO, USA) were diluted in 0.1 mM Tris–HCl (pH 7.4) and 0.15 M NaCl buffer and carried to a final volume of 600 µL. After 2.5 h of incubation at 37 °C, the reaction was stopped by adding 500 µL of 5% (*w*/*v*) trichloroacetic acid (TCA) at 37 °C for 30 min at room temperature. The mixture was centrifuged at 12,000× *g* for 10 min, and casein hydrolysis was measured from 200 µL of the supernatant at 280 nm. In the second set of tubes, venom samples were incubated with 50 mM EDTA (Sigma-Aldrich, catalog number E9884, St. Louis, MO, USA) for 30 min before casein was added and the same procedure was performed (EDTA inhibits activity of the metalloproteases). The total activity minus the activity in the presence of EDTA represents the activity of metalloproteases. Water instead of venom was used as a negative control and it was subtracted from the venom samples. The specific activity is reported as activity units (AUs) per microgram of AU)/venom protein, where one activity unit corresponded to an increase in absorbance of 0.01 at 280 nm.

#### 2.5.3. Phospholipase A_2_ (PLA_2_) Activity

The activity of phospholipase A_2_ of 100 ng of venom protein was analyzed by a colorimetric assay using an sPLA_2_ Assay Kit (Cayman Chemical, catalogue number 765,001, Ann Arbor, MI, USA). The assay used diheptanoyl phosphatidylcholine analog as the substrate. After hydrolysis of the substrate at the SN-2 position of the thioester bond by the phospholipases, the free thiols generated were detected using 5,5′-dithio-bis-(2-nitrobenzoic acid) (DTNB). The absorbance increase was measured using a spectrophotometer (Benchmark Plus, Bio-Rad, USA) at 414 nm each min for 10 min. The bee venom PLA_2_ (Cayman Chemical, catalogue number 765,016, Ann Arbor, MI, USA) was used as the positive control. The PLA_2_-specific activity is expressed as hydrolyzed substrate per minute per microgram of protein. Water instead of venom was used as a negative control and was subtracted from the venom samples.

#### 2.5.4. Hyaluronidase Activity

Concentrations of 20–200 μg of venom protein were used. A sample of 100 μL of buffer and 100 μL of 1 mg/mL of hyaluronic acid (Sigma-Aldrich, catalog number H7630, St. Louis, MO, USA) as substrate were added to a final volume of 250 μL (buffer volume varies depending on the volume of each sample concentration to be used). After mixing, it was incubated for 15 min at 37 °C, and subsequently, 1 mL of 2.5% hexadecyltrimethylammonium bromide in 2% NaOH was added and allowed to stand for 30 min, which was read at a wavelength of 400 nm in a plate reader (Benchmark Plus, Bio-Rad, USA) [38].

### 2.6. RP-HPLC

The analysis was conducted using 200 μg of the adult pooled venoms of MCa-PV and FCp-PV and the individual hybrid samples MH1, MH4, MH6, MH7, and FH3 using a C18 column (150 × 4.6 mm; particle size: 3.5 μm) equilibrated with solution A (0.1% trifluoroacetic acid TFA in H_2_O) using an Agilent 1200 chromatograph (Agilent 1200, Agilent Technologies, CA, USA). Elution was performed at 1 mL/min using a gradient toward solution B (acetonitrile ACN and 0.1% of TFA) as follows: 0% B by 5 min, 0–60% B over 60 min. Absorbance was monitored at 280 nm.

### 2.7. MALDI-TOF-MS

The adult pooled venom samples of MCa-PV and FCp-PV and the hybrids (MH1, MH4, MH6, MH7, and FH3) were analyzed using 70 µg diluted in 3 µL of sinapinic acid matrix (10 mg/mL in ddH_2_O), spotted onto MALDI target plates and analyzed using an Ultraflex-TOF/TOF mass spectrometer (Bruker Daltonics, Bremen, Germany). The mass analyzer was operated in reflector mode using a 20 kV accelerating voltage to measure the ion abundance. The software Flexcontrol (Bruker Daltonics, Bremen, Germany) was used for data analysis.

### 2.8. Acute Toxicological Assay 

The LD_50_ of the pooled venoms of four male hybrids (MH-PV; MH1, MH4, MH6, and MH7), three female hybrids (FH-PV; FH2, FH3, and FH5), Cabf, and Cpbm was measured in groups of four mice of the CD-1 strain (18–20 g) by intraperitoneal injection of 0.5 mL of venom doses (0.5, 0.75, 1, 1.5, 2, and 3 μg of venom protein per milliliter of saline water), and a control group injected with isotonic saline solution was included. The venom was kept at −70 °C and was used only during the first three days after milking. The animals were kept under controlled temperature, humidity, and 24 h light-dark cycle conditions. The LD_50_ was calculated by linear regression analysis of the log_10_ of the doses versus the percentage of survival within the first 24 h of venom injection. The protocol was approved by the Bioethics Committee of the Natural Sciences Faculty of the Autonomus University of Querétaro, México (01FCN2014). At all times, the ethical guidelines of the Norma Oficial Mexicana NOM-062-ZOO [39] were observed. 

## 3. Results and Discussion

### 3.1. Phenetic Analysis

*C. polystictus* appears in the triseriatus group, a small montane rattlesnake species [40], where *C. aquilus* is included. In analyses (by BEAST) based on gene sequences of mitochondrial DNA, it appears as a clade sister to the *C. durissus* group; however, analysis by BI (concatenated Bayesian) groups it as a sister of the *C. cerastes* group and as a sister group of the *C. triseriatus* group in ML (maximum likelihood) analysis conducted by Reyes-Velazco et al. [41] and by Alencar et al. [22]. This variation indicates that more analyses are needed to establish their phylogenetic relation.

To study the phenetic relationship between *C. aquilus*, *C. polystictus*, and the hybrids, the characteristics of the number of scales and two protein bands were used. With respect to scales, the hybrids showed a combination of numbers and types from both parental species, according to data reported by Meik et al. [20] for *C. aquilus*, and by McCranie [18] for *C. polystictus* (Table 1). The hybrids presented a mixture of dorsal coloring patterns from the parents, with similarities to one or the other parent. MH1 was similar to Cabf, whereas MH6 was similar to Cpbm (Figure 1A–D).

With respect to the venom electrophoretic profiles, some differences in the protein banding patterns were observed. Cpbm presented four protein bands around 28–34 kDa, while Cabf presented only two. The protein pattern between the hybrids MH1, FH2, FH3, MH4, and MH6 and Cpbm was similar, showing a protein band of an apparent molecular weight of 31 kDa (Figure 1E). On the contrary, the protein pattern of FH5 and MH7 was more similar to that of Cabf, presenting a protein band with an apparent molecular weight of ~26 kDa, also present in the other hybrid venoms but not in Cpbm. The hybrids had a combination of protein bands from both parental species, including ~31 and ~26 kDa bands, which were present in most of the hybrids (MH1, FH2, FH3, MH4, and MH6) and apparently not shared among the biological parents. FH5, MH6, and MH7 (second hybridization event), unlike the other hybrids, showed a greater number of low molecular weight bands—less than 7 kDa. 

In the phenogram obtained by using the characteristics of the number of scales and two protein bands of ~26 and ~31 kDa specific for the parental Cabf and Cpbm venoms, two main groups were identified. The first was conformed to by all of the hybrids and the *C. polystictus* adults (MCp10 and Cpbm). The second group was formed by the *C. aquilus* adults (MCa8, MCa9, and Cabf). FH5 was more similar to the biological mother (Cpbm) than to *C. aquilus* (Figure 2A). However, it is important to observe that even though MH7 and Cabf were clearly similar in their protein patterns, they were divided into different groups in the cluster analysis. This could be due to the cluster analysis being conducted using only the following two differential protein bands: Cabf (26 kDa) and Cpbm (31 kDa), which were not shared between them. All of the other protein bands were not included in the analysis due to being shared between the parental venoms.

The phylogenetic analyses using the morphological characteristics (scales and gene sequences) [41,42] showed results indicative of high similarity between *C. aquilus* and *C. polystictus*, and the hybridization events indicate that the reproductive isolation between *C. aquilus* and *C. polystictus* is not complete [43] and that they could be closely related. In addition, their distribution converged in central Mexico, where, in some areas, *C. aquilus* and *C. polystictus* are sympatrically distributed [18,19], which could represent a possible natural hybridization zone (Figure 2B). 

### 3.2. Enzymatic Activity and Zymography

#### 3.2.1. Snake Venom Serine Proteases (SVSPs)

The zymography for both serine protease types showed bands in the range of ~20 to ~90 kDa for the hybrids and were similar to the previously reported protein pattern banding of MCa-PV and FCp-PV [44]. In trypsin-like zymography, the hybrids MH1, MH4, and MH6 showed a similar band pattern to Ca, presenting a protease band of ~90 kDa, while the hybrids FH3 and MH7 and FCp-PV lacked this protease band (Figure 3A,B). A high molecular weight chymotrypsin-like activity was detected by zymography with a band of ~90 kDa that was present only in the hybrids MH1 and MH4 and in MCa-PV. FH3 showed a protease of ~35 kDa, while MH6 presented a protease of ~85 kDa, both of which were not detected in MCa-PV, FCp-PV, or the other hybrids. MH7 and Cp showed no chymotrypsin-like activity by zymography (Figure 3C,D). For both types of zymography, the hybrids MH1 and MH4 shared a protease band of ~90 kDa, not shown in *C. polystictus*, perhaps inherited from *C. aquilus.*

For trypsin-like enzymatic activity, both MCa8 and MCp10 (excluding Cabf with lower activity) showed similar activities. Cpbm showed the highest trypsin-like activity between the parental species. In terms of the chymotrypsin-like activity, the organisms of *C. aquilus* showed the highest activity compared to the *C. polystictus* organisms. The hybrids FH2 and MH7 showed the highest trypsin-like enzymatic activity, whereas, for chymotrypsin-like activity, this was presented by MH1. For the parental species, the highest value of elastase-like activity corresponded to FCp-PV, while both the enzymatic activities between the hybrids MCa-PV and FCp-PV were similar. The highest values of enzymatic activity between the serine proteases corresponded to trypsin-like proteases (Figure 4).

The proteolytic activity in the solution, as well as the zymography, showed high variability between the hybrids and between the adults. However, in terms of the enzymatic values of the trypsin-, chymotrypsin-, and elastase-like activities, they were similar or higher for some hybrids’ venoms compared to their biological parents’ venoms. Hybrids from *Bothrops erythromelas* and *Bothrops neuwiedi* showed a high variability in their enzymatic activity, particularly in neonates, where an increase in trypsin-like activity was related to an increase in body growth, with similar activity to the parents when the hybrids were 12 months old [11]. Here, we observed that the three- and four-year-old hybrids showed similar values of trypsin- and chymotrypsin-like proteases and, in some cases, higher enzymatic activity than their parents. 

#### 3.2.2. Gelatinolytic Activity

After SDS-PAGE, the snake venom metalloproteases (SVMPs) were identified using gelatin as the substrate, copolymerized with polyacrylamide, and incubated for 24 h (Figure 5A,B). A similar proteolytic banding pattern was observed with proteases of ~200, ~86, ~60, ~50, and ~30 kDa between the hybrids and adults. These data are similar to those reported for *C. aquilus* and *C. polystictus* by this work group [44]. The hybrids MH1, FH2, FH5, MH6, and MH7 and Cpbm, MCa8, and MCa9 presented a band of ~86 kDa MW, not observed in Cabf. The hybrids MH1, FH2, and FH3 and the *C. polystictus* adults did not present the ~31 kDa protease present in the other organisms. Moreover, a band of ~50 kDa was observed in all of the organisms, but only the hybrids and *C. aquilus* individuals showed proteolytic activity. The highest metalloprotease activity was present in MCa-PV and FCp-PV venoms and the other hybrids, which showed similar values of enzymatic activity (Figure 5C).

#### 3.2.3. Snake Venom Hyaluronidases (SVHs)

The hyaluronidase activity was determined by zymography and turbidimetry using hyaluronic acid (Figure 6). The venoms of the parents showed hyaluronidases of ~12, ~25, and ~50 kDa (absent in MCa-PV), as previously reported [44], and these protein bands were found in all of the hybrid samples studied. The turbidimetric assay showed a lower activity for Ca than the hybrids MH1, MH4, and MH7 and Cp. More variability in the hyaluronidase pattern banding was observed for the ~25 to ~50 kDa bands; for example, MCa-PV did not present hyaluronidases in the range of ~50 kDa. Cp showed higher values of SVH than MCa-PV, while the other hybrids showed similar activity values to MCa-PV but lower than FCp-PV. The enzymatic values of the SVHs were higher for the MH7 and MH4 hybrids compared to the other hybrids and to MCa-PV and FCp-PV. The data were similar to those reported; however, SVH proteins of ~90 were not detected [44]. 

#### 3.2.4. Snake Venom Phospholipase A_2_ (SVPLA_2_) Activity

For phospholipase A_2_ activity, the zymography showed a phospholipase band of ~14 kDa in all of the samples, similar to the reported data [44] (Figure 7); however, MCa-PV and the hybrid FH3 showed the highest activity. The MH6 showed higher phospholipase A_2_ activity than Cp, while the other hybrids showed similar/or lower values to the MCa-PV and FCp-PV species. The variability in the enzymatic activity values could indicate the presence of different isoforms of PLA_2_ in the analyzed venoms, which may be related to inter- or intraspecific variation in the venoms [45].

### 3.3. RP-HPLC and MALDI-TOF-MS

The protein fractionation by RP-HPLC showed different profiles between the venoms. Peaks between 35 and 55 min were found, where proteins such as phospholipase A_2_ (35–45 min) [46] and metalloproteases and hyaluronidases (≥50 min) [47] are usually eluted. Fewer peaks were found between 20 and 30 min retention times, in accordance with serine proteases (30–35 min) [48]. The last group, probably small peptides, eluted between 5 and 10 min [47]. The reported data for MCa-PV and FCp-PV showed eluted peaks at 36 min, which were not shared between them [44]. These peaks were present in the venoms of some of the hybrids, with a greater variability at 25 and 30–45 min, corresponding to the elution times of SVSP and SVPLA_2_; fractions were not detected after 60 min (Figure 8). However, further identification of the proteins must be confirmed by other methods. 

With respect to MS (Figure 9), the hybrids showed lower peak intensity for SVMP compared to the MCa-PV and FCp-PV venoms. These results are consistent with the SVMP enzymatic activity. The peaks of lower molecular weight and in the range of SVPLA_2_ were the most abundant in the hybrids, whereas the peaks in the range of SVSP and SVMP were the most abundant for the MCa-PV and FCp-PV venoms. The FH3 venom showed a greater abundance of peaks, followed by the MH, MCa-PV, and FCp-PV venoms. Additionally, all of the venoms presented compounds of lower molecular weight (3–11 kDa), suggesting crotamine-like myotoxins (MYO)—a small, non-enzymatic protein of approximately 4–5 kDa with a spastic paralysis effect in the hind limbs of mice and necrosis in muscle cells [49]. These MYO peaks were more abundant and variable in the hybrid organisms. Peaks not shared between the parental venoms were observed. The hybrids, either male or female, also presented peaks not shared between them or between their parents. The venom profiles indicate a differential pattern of protein between the parental and hybrid venoms (Table 2). The molecular weights of SVSP, SVMP, and SVPLA_2_ were similar to those observed by zymography; however, a larger number of proteases were found by zymography than by MALDI-TOF-MS, perhaps due to the long incubation time used to measure the activity.

**Table 2 biology-11-00661-t002:** Enzyme families observed for *C. aquilus*, *C. polystictus*, and the hybrids.

Protein Family	Data Reported	MCa-PV	FCp-PV	MH	FH
SVMP	Reported for *C. polistictus* in the range of 20 and 30–40 kDa [50,51]	23.2 kDa	23 kDa	23 kDa	23.4, 37.8, and 46.2 kDa
SVSP	Reported for *C. molossus* venom between 24 and 30 kDa [49]	27 kDa	24.7 and 29 kDa	24.7 kDa	NF
SVPLA_2_	Reported in the range of 13–18 kDa [50]	12.4, 16.2, and 13.9 kDa	11.5, 12.4, 13.9, and 16.2 kDa	11.5, 12.3, and 13.6 kDa	11.6, 13.9, and 17.6 kDa
MYO	Reported in the range of 4–5 kDa in snake venoms and one MYO of 10 kDa in *C. molossus* [49]	6.9 and 7.7 kDa	7.4 kDa	3.7, 4.8, 6.8, and 7.4 kDa	3.7, 5.8, 6.9, 7.7, 8.7, and 9.7 kDa

MCa-PV, FCp-PV, four male hybrids (MH1, MH4, MH6, and MH7) (MHs) and a female hybrid (FH3). NF, not found.

### 3.4. Lethal Dose-50 (LD_50_)

In order to calculate the LD_50_, concentrations of 0.5, 0.75, 1, 1.5, 2, and 3 μg of venom protein per milliliter in saline water were tested. Pooled venoms were used because of the low amount of venom from the juvenile organisms. Pool 1, which comprised venom from the female hybrids (FH-PV; FH2, FH3, and FH5), pool 2, which comprised venom from the male hybrids (MH-PV; MH1, MH4, MH6, and MH7), and the individual samples of the biological father, *C. aquilus* (Cabf), and the biological mother, *C. polystictus* (Cpbm), were tested. The FH-PV and Cabf venoms showed a lethal effect as a function of concentration, with an LD_50_ of 1.02 and 2.04 μg of venom protein per gram of body weight, respectively. A similar LD_50_ value of 2.24 μg of venom protein per gram of body weight was reported for *C. aquilus* [52]. For MH-PV and Cpbm, no lethal effects were detected with the doses used; however, it has been reported to have an LD_50_ of 3.4 μg of venom per gram of body weight [53], of 4.5 μg of venom protein per gram of body weight in neonate venom, and of 5.5 μg of venom protein per gram of body weight in adult venom of *C. polystictus* using non-Swiss albino mice [54]. Hybridism has been shown to be associated with variability in venom composition [2,11], and in previous reports, high variability in enzymatic activity has been observed [11]. In this study, the female hybrids showed the lowest LD_50_, meaning that they have a more toxic venom than their parents. This could be associated with the high variability of enzymes observed as a result of genetic variability due to the toxic effect of the venom being a result of the combination or synergism of the different enzymes [3].

With the exception of metalloprotease activity, some hybrids showed similar or higher enzymatic values (MH1, FH2, FH3, and MH7) with respect to the adults (Cabf, Cpbm, MCa-PV, and FCp-PV). When comparing the venoms of the juvenile hybrids (trypsin- and chymotrypsin-like activities) or the adult hybrids (elastase-like, SVPLA_2_, and SVHA activities) versus the parents, it was of particular interest that FH3, included in the pooled venom with the lowest LD_50_, showed similar or higher values in both SVSP (trypsin-, chymotrypsin-, and elastase-like protease) and PLA_2_ activities compared to the other organisms. However, despite male hybrids in some cases showing higher values of SVSP (trypsin-like) or similar PLA_2_ activities than some of the female hybrids, none of the male hybrids showed high enzymatic activity. For example, MH1 and MH7 showed similar or higher SVSP (trypsin-, chymotrypsin-, and elastase-like) proteolytic activity compared to the female hybrids and Cabf and Cpbm, but they showed lower PLA_2_ activity than MCa-PV and FH3 (which presented lethality under the used doses). The PLA_2_ activity for the MH1, MH4, and MH7 hybrids was similar to that of *C. polystictus* (which did not present lethality under the used doses). Neither parental species showed both high SVSP and PLA_2_ activities compared to the female hybrids. Moreover, it was observed that the male hybrid MH6 showed an SVPLA_2_ activity similar to the female hybrid FH3, but lower SVSP (chymotrypsin-like) and trypsin-like activities than the female hybrid FH2. For *C. molossus nigrescens* venom, it has been reported that at least two SVSPs are the main compounds associated with its lethality [49]. Additionally, PLA_2_ is one of the main components of snake venom, which helps immobilization and rapid death of prey and is associated with myotoxic, cytolytic, neurotoxic, and edema formation [44,51]. Refs. [49,55] Perhaps high values of both families of enzymes could be related to the lethality of the female hybrids that showed high SVPLA_2_ and SVSP activity values. These results suggest an important role for SVPLA_2_, SVSP, and MYO in the lethality of hybrids. 

In other species, such as *Crotalus viridis* or *Crotalus horridus*, whose venoms are usually hemotoxic, neurotoxins have also been found and explained, based on their hybridization with *Crotalus* species, which are phylogenetically related, with the presence of neurotoxins in their venoms. The presence of neurotoxins in hybrids is assumed to be an adaptive advantage and a source of intraspecific variation among snake venom; however, it has been reported that in the hybridization of *C. s. scutulatus* and *C. viridis*, the Mojave toxin (PLA_2_) is not necessarily a strong selective advantage source for snake venom due to the fact that the toxins acquired through this phenomenon are confined to specific areas or hybridization zones where the two species are in near contact and the toxins acquired do not spread outside of the hybridization boundaries into the gene pool of the parental populations. However, the authors cannot exclude the possibility of a different outcome under different selective regimes, since they cannot reject the possibility that, given sufficient time, a degree of introgression of Mojave toxin genes may occur [3]. Moreover, ontogenetic changes are related to a low LD_50_, as some species of rattlesnakes as neonates could have more lethal venom than adults, which tends to show higher SVMP enzymatic activity, as observed in studies in adults and neonates of C. *polystictus*. However, no significant differences between the juveniles’ (PLA2, SVMP, and kallikrein-like) and adults’ enzymatic activity were observed [54]. Santoro et al. [11] reported ontogenetic changes (lower amidolytic and proteolytic activities) in the neonate hybrids of *Bothrops erythromelas* and *Bothrops neuwiedi* when they were three months old, with no differences to their parents when they were 12 months old. The analyses in this study showed juvenile hybrids of three and four years old with low SVMP activity and more protein, peptide, and enzyme variability in the protein families of their venoms than the parental species. This variation could include several isoforms that may present a distinct functional activity and could represent an advantage in the functionality, improving the capture of prey, digestion, or the defensive efficacy of the venom [55,56,57] and could have adaptive implications, facilitating the response to changing environmental pressures [58,59]. 

## 4. Conclusions

The phenetic analysis indicated a higher similarity between the hybrids and *C. polystictus* than with *C. aquilus*; however, the lethality of their venom was different, with the venom of the juvenile female hybrids being more lethal than that of their corresponding parents. Snake venom is a very complex mixture of different compounds, and its lethality is mainly the result of a combination of enzymes and toxins. The high values in SVSP and PLA_2_ and the higher variability of peptides of low molecular weight (maybe crotamine-like peptides) in the female hybrids could be related to their toxicity. The phenomenon of snake hybridization represents a way to increase the variability of snake venoms and an opportunity to observe changes in some specific protein families, which could help us to understand the mechanism of variability itself and in the development of more effective antidotes. 

## Figures and Tables

**Figure 1 biology-11-00661-f001:**
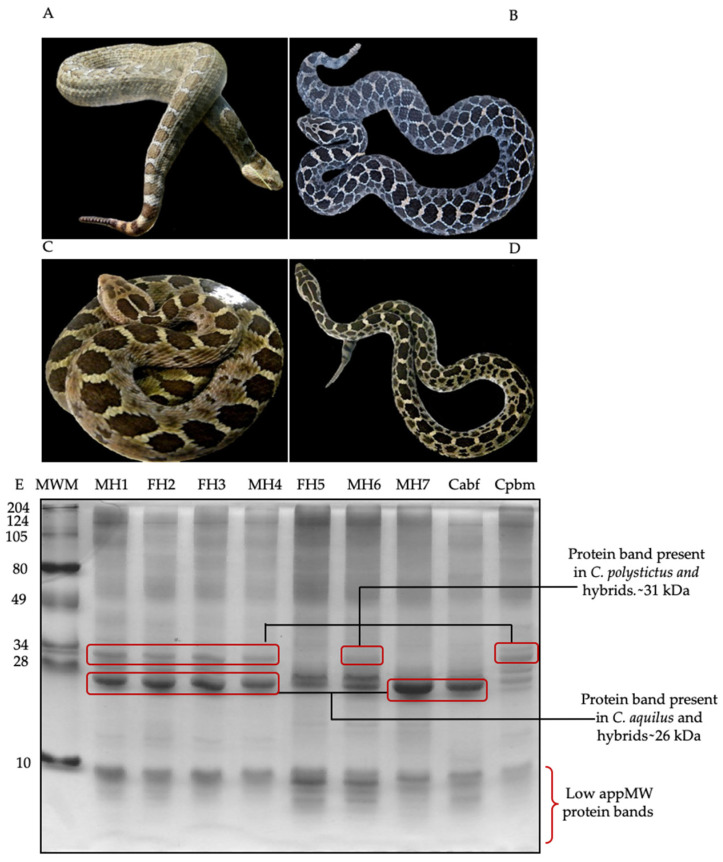
Dorsal banding pattern and protein profile of 20 µg of venom in a 12% SDS-PAGE. (**A**) *C. aquilus* (Cabf), (**B**) *C. polystictus* (Cpbm), (**C**) male hybrid 1 (MH1), (**D**) male hybrid 6 (MH6), (**E**) protein pattern. Hybrids of the first hybridization event (four years old): MH1, FH2, FH3, and MH4; hybrids of the second hybridization event (three years old): FH5, MH6, and MH7; *C. aquilus* biological father (Caqbf); *C. polystictus* biological mother (Cpbm). Molecular weight markers (MWMs) are indicated on the left side. M, male; F, female.

**Figure 2 biology-11-00661-f002:**
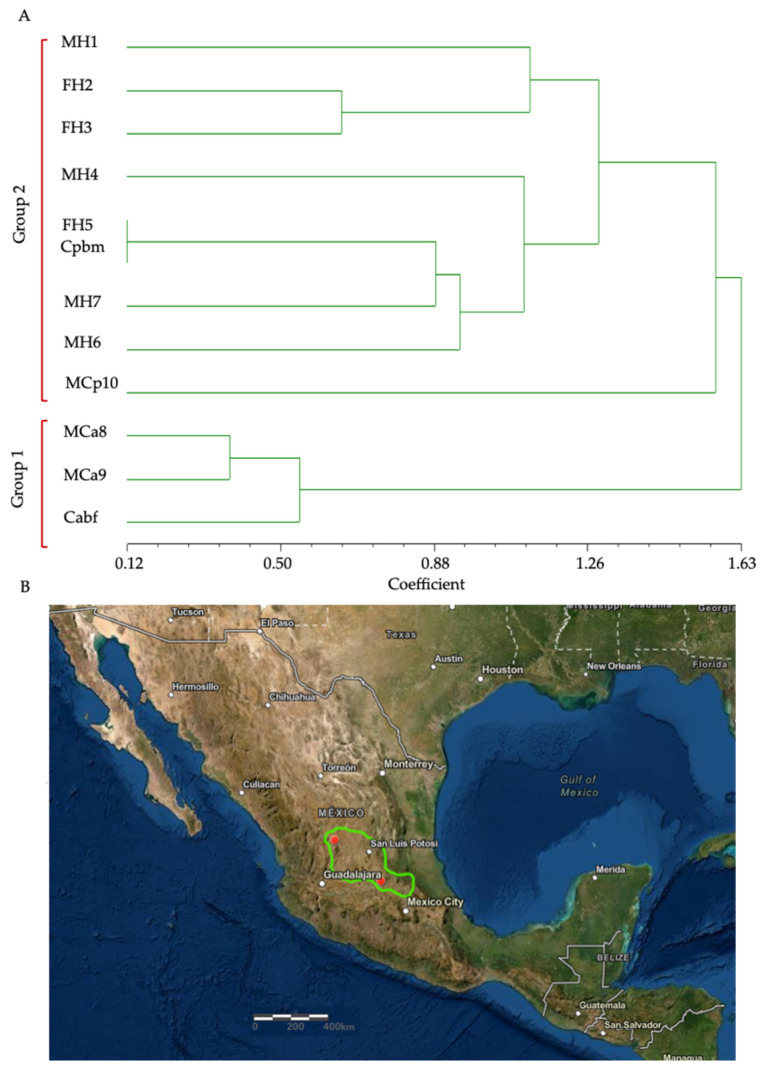
Phenogram of *C. aquilus*, *C. polystictus*, and hybrid rattlesnakes. The phenetic diagram was constructed using the UPGMA method [27] based on the number of scales and protein bands. (**A**) Hybrids of the first hybridization event (four years old): MH1, FH2, FH3, and MH4; hybrids of the second hybridization event (three years old): FH5, MH6, and MH7; males of *C. aquilus* (MCa8 and MCa9) and *C. aquilus* biological father (Cabf); *C. polystictus* male (MCp10) and *C. polystictus* biological mother (Cpbm). M, male; F, female. (**B**) possible sympatric distribution of *C. aquilus* and *C. Polystictus* in central Mexico.

**Figure 3 biology-11-00661-f003:**
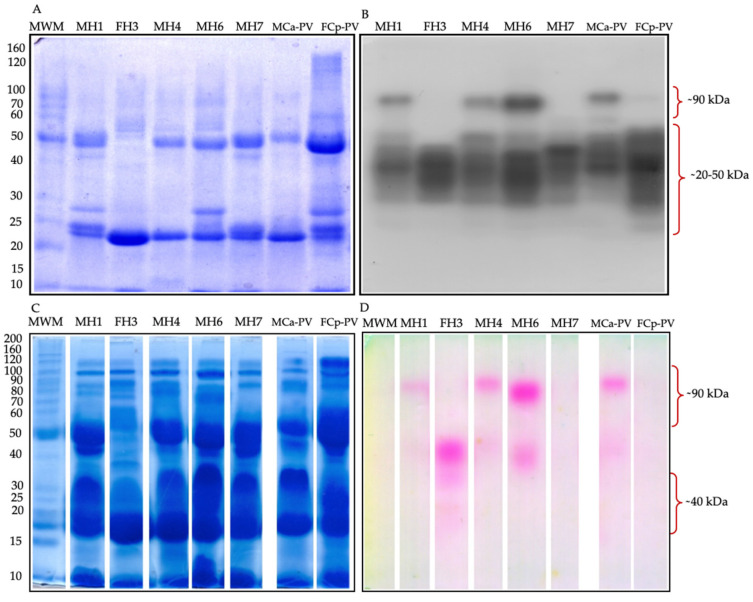
Venom serine protease-like zymography. (**A**) 50 µg of venom protein in a 10% SDS-PAGE gel. (**B**) Trypsin-like activity was determined using BApNA as the substrate and incubated for 2.5 h. (**C**) 100 µg of venom protein in a 10% SDS-PAGE gel. (**D**) Chymotrypsin-like proteolytic activity determined using SAAFpNA as the substrate and incubated for 2.5 h. Hybrids of the first hybridization event (11 years old): MH1, FH3, and MH4; hybrids of the second hybridization event (10 years old): MH6 and MH7; pooled venom of four male organisms of *C. aquilus* (MCa-PV), pooled venom of four female organisms of *C. polystictus* (FCp-PV). MWM, molecular weight marker.

**Figure 4 biology-11-00661-f004:**
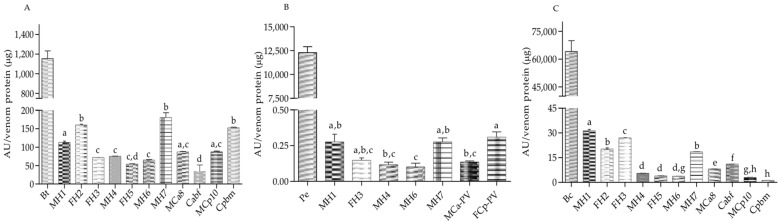
Specific enzymatic activity of serine proteases. (**A**) Trypsin-like specific activity on BApNA substrate; (**B**) chymotrypsin-like specific activity on SAAPFpNA substrate; (**C**) elastase activity of 5 µg of venom protein. Hybrids of the first hybridization event (four years old): MH1, MH2, FH3, and MH4; hybrids of the second hybridization event (three years old): FH5, MH6, and MH7; males of *C. aquilus*: MCa8 and Cabf; organism of *C. polystictus*: A male, MCp10, and a female, Cpbm, used for the determination of trypsin- and chymotrypsin-like activities. Hybrids of the first hybridization event (11 years old): MH1, FH3, and MH4; hybrids of the second hybridization event (10 years old): MH6 and MH7; and pooled venom of *C. aquilus* (MCa-PV) and of *C. polystictus* (FCp-PV) were used for the determination of elastase-like activity. Bt (bovine trypsin), Bc (bovine chymotrypsin), and Pe (porcine elastase) were used as positive controls. Lowercase letters indicate significant statistical differences (Tukey; *p* < 0.05).

**Figure 5 biology-11-00661-f005:**
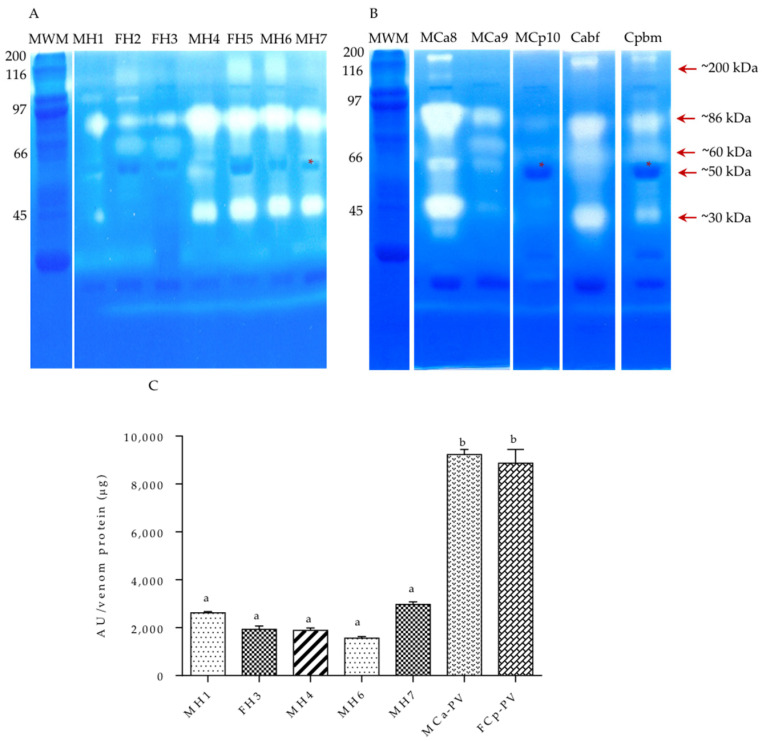
Venom metalloproteinase activity. (**A**) Zymography in SDS-PAGE co-polymerized with gelatin using 20 µg of protein venom under no reducing conditions. (**B**) Enzymatic activity of 100 µg of venom protein. Hybrids of the first hybridization event (four years old): MH1, MH2, FH3, and MH4; hybrids of the second hybridization event (three years old): FH5, MH6, and MH7; males of *C. aquilus*: MCa8, MCa9, and Cabf; organism of *C. polystictus*: A male, MCp10, and a female, Cpbm, used for zymography analysis. (**C**) Hybrids of the first hybridization event (11 years old): MH1, FH3, and MH4; hybrids of the second hybridization event (10 years old): MH6 and MH7; and pooled venom of venoms MCa-PV and FCp-PV were used for the determination of metalloproteinase activity. MWM, molecular weight marker. Red asterisk shows a protein shared between *C. polystictus* and the hybrids. Lowercase letters indicate significant statistical difference (Tukey; *p* < 0.05).

**Figure 6 biology-11-00661-f006:**
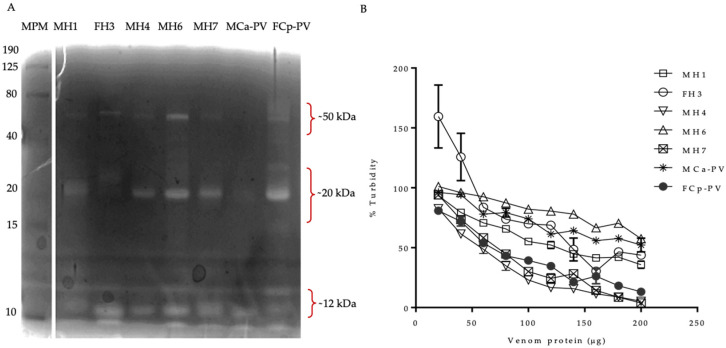
Hyaluronidase activity. (**A**) Hyaluronidase zymography of 100 µg of venom protein. (**B**) Enzymatic activity of hyaluronidases using 20–200 µg of venom protein. Hybrids of the first hybridization event (11 years old): MH1, FH3, and MH4; hybrids of the second hybridization event (10 years old): MH6 and MH7; pooled venom of MCa-PV and FCp-PV. The MWMs are to the left.

**Figure 7 biology-11-00661-f007:**
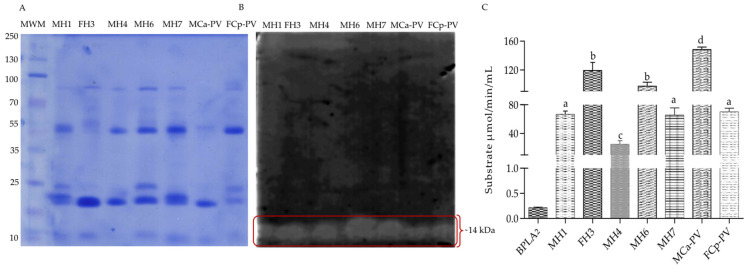
Snake venom phospholipase A_2_ activity. (**A**) 10% SDS-PAGE using 100 ng of venom protein. (**B**) Zymography on agarose gel copolymerized with egg yolk. (**C**) Enzymatic activity of 100 ng of venom protein. Hybrids of the first hybridization event (11 years old): MH1, FH3, and MH4; hybrids of the second hybridization event (10 years old): MH6 and MH7; pooled venom of MCa-PV and FCp-PV. The MWMs are to the left. Lowercase letters indicate significant statistical difference (Tukey; *p* < 0.05).

**Figure 8 biology-11-00661-f008:**
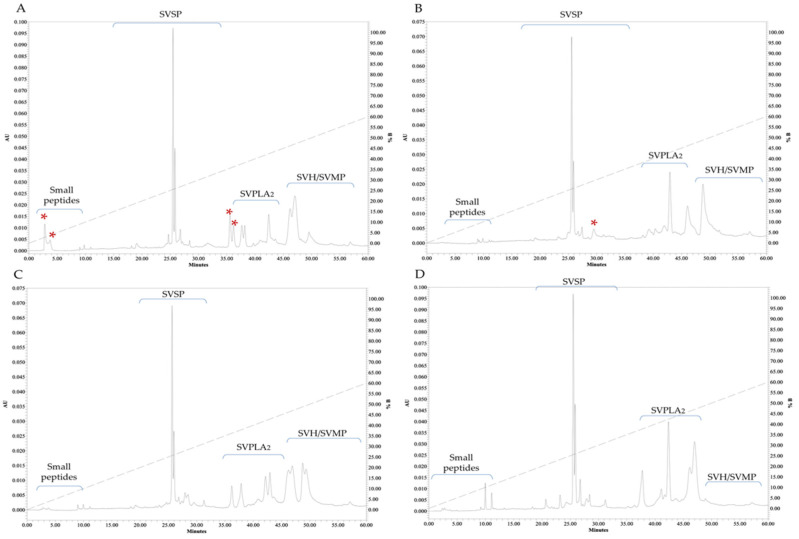
Protein fractionation of the venom by RP-HPLC. Pooled venoms of *C. aquilus* (**A**), *C. polystictus* (**B**), the male hybrids (MH1, MH4, MH6, and MH7) (MHs) (**C**), and a female hybrid (FH3) (**D**). Red asterisks show the peaks not shared between the samples. SVSP, snake venom serine protease; SVMPs, snake venom metalloprotease; SVPLA_2_, snake venom phospholipase; SVH, snake venom hyaluronidase. Absorbance units (AUs) at 280 nm. Enzymatic activity was not determined. The family proposed for the enzymes was based on similarity of the elution profiles with the reported data.

**Figure 9 biology-11-00661-f009:**
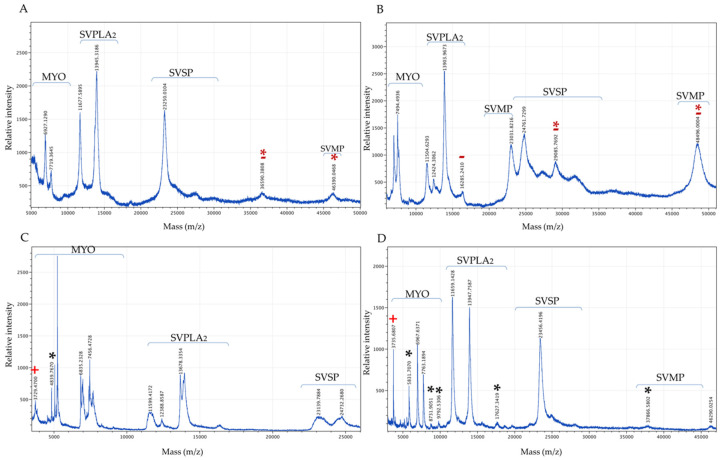
MALDI-TOF-MS analysis of snake venom. Pooled venoms of MCa-PV (**A**), FCp-PV (**B**), the male hybrids (MH1, MH4, MH6, and MH7) (MHs) (**C**), and a female hybrid (FH3) (**D**). Red asterisks indicate the peaks not shared between the MCa-PV and FCp-PV venoms; red minus signs indicate exclusive peaks for MCa-PV and FCp-PV; black asterisks indicate exclusive and unshared peaks between the male and female hybrid venoms; red plus signs indicate a peak present only in both hybrid venoms. MYO, myotoxin; SVSP, snake venom serine protease; SVPLA_2_, snake venom phospholipase; SVMP, snake venom metalloprotease. SVPLA_2_ and SVSP peaks.

**Table 1 biology-11-00661-t001:** Comparison between number of scales for *C. aquilus*, *C. polystictus*, Cabf, Cpbm, and the hybrids.

Scales	MCa+	FCa+	Cabf	MCp *	FCp *	Cpbm	MHs	FHs
Dorsal spotting pattern	24–35	26–32	27–30	30–47	30–47	ND	32–39	31–34
Tail band pattern	2–4	4–6	4	ND	ND	5	4–7	3–5
Scales around the rattle	8–10	8–10	10	ND	ND	12	10–12	10–12
Subcaudal	18–24	23–28	25–26	24–29	17–25	20	24–27	19–21
Ventral	141–150	136–144	143–152	161–177	167–187	165	155–158	157–164
Supralabial	11–13	10–13	11–12	12–15	12–15	14	13–14	12–14
Infralabial	11–12	10–12	11–12	11–16	11–16	14	11–14	12–14
Interrectal	20–25	20–26	ND	ND	ND	ND	ND	ND
Middle body scales	21–24	21–23	23	25–28	25–28	25	25–26	25
Intersupraocular	3	2–4	ND	ND	ND	3	ND	ND

Data of the number of scales of *C. aquilus*: Males (MCa+) and females (FCa+) [15] and the *C. aquilus* biological father (Cabf); *C. polystictus*: Males (MCp *) and females (FCp *) [18], the *C. polystictus* biological mother (Cpbm), four male hybrids (MHs), and three female hybrids (FHs). * Not determined (ND).

## Data Availability

This manuscript was read and approved by all of the named authors, and there were no other persons who satisfied the criteria for authorship but are not listed. The order of authors listed in the manuscript was approved by all authors. The authors gave due consideration to the protection of intellectual property associated with this work, and there are no impediments to publication, including the timing of publication, with respect to intellectual property. The authors followed the regulations of their institutions concerning intellectual property. Any aspect of the work covered in this manuscript that involved experimental animals was conducted with the ethical approval of all relevant bodies, and such approvals are acknowledged within the manuscript. The corresponding author is the sole contact for the editorial process (including the editorial manager and direct communications with the office). They are responsible for communicating with the other authors about progress, submissions of revisions, and final approval of proofs. Current, correct email addresses from all authors have been provided, accessible by the corresponding author, and configured to accept emails.

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
