# Peer review of "Hybridization between Crotalus aquilus and Crotalus polystictus Species: A Comparison of Their Venom Toxicity and Enzymatic Activities"

_biology, 2022, doi:10.3390/biology11050661_

Round 1

Reviewer 1 Report

The manuscript analyzes the morphology and the venoms of hybrid snakes which resulted from the inbreeding of rattlesnakes Crotalus aquilus and C. polystictus from Mexico. They used the dorsal spotting and tail bands patterns, scales number, and venom protein bands in PAGE-SDS were analyzed to construct a phenogram. The venoms were analyzed concerning SVSP, SVMP, SVPLA2, and hyaluronidase enzymatic activities. The venom lethality was determined, and the profiles resulting from RP-HPLC and MALDI-TOF-MS were analyzed. These characteristics were analyzed with those presented by the parental specimens.

The manuscript is clear and concise, and it has important data about the venoms of C. aquilus, C. polystictus, and the hybrids rattlesnakes. The authors also discuss the species concept between these related species that share the habitat as also venom variability. The data presented show the hybridization among both species in morphology and venom activities.

Remarks

In order to facilitate the reading, it is important to use a unique name to venoms and venom pools to all over the text.

For example:

In Material and Methods 2.6 and 2.7, change the name H1m to Mh1, as used throughout the text.

I had difficulty knowing if Cp is a venom pool of female snakes as indicated in figure 3, or Cp is a venom pool of male and female snakes because there is no reference of gender in other figures. It is interesting to include MCa and FCp in line 108.

Suggestion: as indicated in fig 2, use MCp to venom pool from a male rattlesnake, in fig. 3; it is interesting to use FCp to venom pool from a female rattlesnake, and if it is a pool with a mixed-gender could be only Cp.

Line 55: suggestion: use Crotalus atrox x C. scutulatus scutulatus [10], Bothrops erythromelas x B. neuwiedi [3], C. s. scutulatus x C. oreganus helleri [11], C. viridis viridis x C. s. scutulatus [7], and Protobothrops flavoridis x P. elegans [12] hybrids. After the first indication, it is possible to abbreviate the genus all over the manuscript.

Line 112: inform the correct quantity of venom used in SDS-PAGE.

Line 122: suggestion “by Ohlsson et al. [25] and Vinokurov et al. [26].”

Line 139: check the reference 27, it is about serine protease.

Line 141: 0.002 % (v/v) Triton X-100 as also in line 146. In line 147: 0.5 M Tris-HCl.

Line 153: Informing the hyaluronic acid concentration used in the enzymatic assay in Material and Methods.

Line 164: it is not necessary to inform the year of publication.

Line 166: complete the reaction volume of trypsin and chymotrypsin assay, and inform the concentration of elastase-like substrate.

Line 181: correct the number of references.

Line 182: Explain better the reaction conditions to understand the final volume and the venom and casein concentrations.

Line 169-170: inform the concentration of venom sample used in enzymatic assays.

Line 205: inform the concentration of hyaluronic acid used in the reaction.

Line 212 and 220: Indicate the abbreviation of venoms of male C. aquilus (MCa) and female C. polystictus (FCp) and change the abbreviation of hybrids.

Results

It is necessary to correct the title of fig. 1, “20 µg of venom”.

Suggestion to fig. 2 legend: Phenogram of C. aquilus, C. polystictus, and hybrid rattlesnakes. The phenetic diagram was constructed using UPGMA method [22] based on scales number and protein bands.

In fig. 2 (B), it is interesting to indicate the distribution of C. aquilus and C. polystictus using different colors.

About MALDI-TOF-MS analysis of snake venom: more data is necessary to confirm that all the small peptides presented in Fig. 9 and Table 2 are crotamine-like peptides.

It is interesting to add the confidence interval, confidence limit, and statistical significance in DL50 data.

Attention to references, the journal recommends the ACS style guide.

Author Response

Thank you very much for your important observations.

Answers: 

In Material and Methods 2.6 and 2.7, change the name H1m to Mh1, as used throughout the text.

I had difficulty knowing if Cp is a venom pool of female snakes as indicated in figure 3, or Cp is a venom pool of male and female snakes because there is no reference of gender in other figures. It is interesting to include MCa and FCp in line 108.

Suggestion: as indicated in fig 2, use MCp to venom pool from a male rattlesnake, in fig. 3; it is interesting to use FCp to venom pool from a female rattlesnake, and if it is a pool with a mixed-gender could be only Cp.

In order to achieve a better understanding, we have changed and homologated all the group names throughout the text.

Line 55: suggestion: use Crotalus atrox x C. scutulatus scutulatus [10], Bothrops erythromelas x B. neuwiedi[3], C. s. scutulatus x C. oreganus helleri [11], C. viridis viridis x C. s. scutulatus [7], and Protobothrops flavoridis x P. elegans [12] hybrids. After the first indication, it is possible to abbreviate the genus all over the manuscript.

Corrections were done in lines 59 to 65.

Line 112: inform the correct quantity of venom used in SDS-PAGE.

We used 20 micrograms of venom protein, correction was done in line 121.

Line 122: suggestion “by Ohlsson et al. [25] and Vinokurov et al. [26].”

Correction was done in line 131, reference numbers changed to 30 and 31.

Line 139: check the reference 27, it is about serine protease.

The article includes the methodology for the identification of gelatinolytic activities by metalloproteases. Reference number changed for 32 in line 148.

Line 141: 0.002 % (v/v) Triton X-100 as also in line 146. In line 147: 0.5 M Tris-HCl.

Corrections were done in lines 150, 156 and 157.

Line 153: Informing the hyaluronic acid concentration used in the enzymatic assay in Material and Methods.

Concentration was added in line 164.

Line 164: it is not necessary to inform the year of publication.

Correction was done in line 175.

Line 166: complete the reaction volume of trypsin and chymotrypsin assay, and inform the concentration of elastase-like substrate.

Line 169-170: inform the concentration of venom sample used in enzymatic assays.

A more detailed explanation of the methods was added in lines 175-187.

Line 181: correct the number of references.

Corrections were done in line 197.

Line 182: Explain better the reaction conditions to understand the final volume and the venom and casein concentrations.

Corrections were done in lines 198-201 and 204-210.

Line 205: inform the concentration of hyaluronic acid used in the reaction.

Correction was done in line 227.

Line 212 and 220: Indicate the abbreviation of venoms of male C. aquilus (MCa) and female C. polystictus(FCp) and change the abbreviation of hybrids.

Corrections were done in lines 243-244, in fact changes were done throughout the whole text.

Results

It is necessary to correct the title of fig. 1, “20 µg of venom”.

Correction was done in line 299.

Suggestion to fig. 2 legend: Phenogram of C. aquilus, C. polystictus, and hybrid rattlesnakes. The phenetic diagram was constructed using UPGMA method [22] based on scales number and protein bands.

Suggestions were accepted, corrections were done in lines 324-325.

In fig. 2 (B), it is interesting to indicate the distribution of C. aquilus and C. polystictus using different colors.

Thank you, the map represents a general distribution for both C. aquilus and C. polystictus, it is not possible to indicate a differential distribution.

About MALDI-TOF-MS analysis of snake venom: more data is necessary to confirm that all the small peptides presented in Fig. 9 and Table 2 are crotamine-like peptides.

Based in the molecular weight, we are proposing that small peptides possibly are crotamine-like peptides, however further analysis must be done for the identification. Therefore, we only suggest their presence in lines 458 and 566.

It is interesting to add the confidence interval, confidence limit, and statistical significance in DL50 data.

The LD50 was calculated by linear regression of the log10 of the doses vs. the percentage of survival, it was indicated in lines 258-259.

Attention to references, the journal recommends the ACS style guide.

Reference style was revised and corrected.

Reviewer 2 Report

The authors analyzed the protein patterns and enzyme activities of the major hemotoxic enzymes of Crotalus aquilus and Crotalus polystictus hybrids and their parents. However, the composition of snake venom varies widely among individuals, even within the same species, depending on region, nutritional status, age, and the environment. As expected, hybrids also vary widely among individuals, and it is not possible to determine whether the effect is due to hybrid formation or simply individual differences, and no new findings have been obtained.
1. There are many errors in the paper. For example, Mh5 on line 266, 5B on line 365, and DL50 on line 475.
2. Even though Mh7 and Cabf are clearly similar in the protein patterns in Figure 1, they are divided into different groups in the cluster analysis in Figure 2.
3. In Figure 5C, enzyme activity in the presence of EDTA should be subtracted as a blank.
4. Figures 6A and 7A are very difficult to see.
5. In Figure 8, proteins can not be identified by RP-HPLC retention time alone.
6. It is inconsistent that hemotoxic enzyme activity is measured in individual snake venoms, but LD50 is measured in pooled venoms.

Author Response

Thank you very much for your important observations.

Answers: 

The authors analyzed the protein patterns and enzyme activities of the major hemotoxic enzymes of Crotalus aquilus and Crotalus polystictus hybrids and their parents. However, the composition of snake venom varies widely among individuals, even within the same species, depending on region, nutritional status, age, and the environment. As expected, hybrids also vary widely among individuals, and it is not possible to determine whether the effect is due to hybrid formation or simply individual differences, and no new findings have been obtained.

Effectively, the variability in the composition of the venoms is widely recognized, there are different factors that are associated with this variability, including the phenomenon of hybridization. We added some information in the introduction, lines 40-51. Although it is true that the variability is multifactorial, research on the composition of venoms between parents and hybrids can provide important information on the possible genetic influence of the parental species and its possible biological implications and medical applications. In example, the LD50 values obtained from the female hybrids indicated a high lethality, followed by the biological father but no lethality was observed for the biological mother.   

There are many errors in the paper. For example, Mh5 on line 266, 5B on line 365, and DL50 on line 475.

In order to achieve a better understanding, we have changed and homologated all the group names throughout the text.

  1. Even though Mh7 and Cabf are clearly similar in the protein patterns in Figure 1, they are divided into different groups in the cluster analysis in Figure 2.

Thank you. The cluster analysis was done using only the two protein bands of Cabf (26 kDa) and Cpbm (31 kDa) because these protein bands were not shared between the parental individuals. All the other protein bands were not included in the analysis due they were shared between the parental venoms. The observation was included in lines 310-315.

  1. In Figure 5C, enzyme activity in the presence of EDTA should be subtracted as a blank.

In this case, the blank includes all the proteolytic activities in the venom sample, whereas in the presence of EDTA, metalloproteases activity was eliminated. Therefore, the total activity minus the activity in the presence of EDTA represents the activity of metalloproteases. Water instead of venom was used as a negative control and was subtracted from the venom samples. Explanation was included in lines 207-210.

  1. Figures 6A and 7A are very difficult to see.

Figures were improved using only bright and contrast modifications.

  1. In Figure 8, proteins cannot be identified by RP-HPLC retention time alone.

Thank you. We agree with your observation and the peaks will be confirmed in the future by further fractionation. The observation was added in line 450.

  1. It is inconsistent that hemotoxic enzyme activity is measured in individual snake venoms, but LD50 is measured in pooled venoms.

Thank you. LD50 was measured in pooled venoms because of the low amount of venom samples from juvenile organisms. We agree with your observation, we know that comparison between individual organisms might be better than using pooled samples. However, even though the LD50 was measured using pooled venoms, it was possible to observe that the sample from female hybrid snakes exhibited the lower value of LD50, suggesting a higher toxic effect than the samples from male hybrids or parental individuals. Explanation was added in lines 489-494.

Reviewer 3 Report

I consider this article to be very interesting and original in the study of phenomenon of snake hybridization and variability of snake venoms. Those finding will result to important medical implications. In further studies, I suggest to add evaluation acetylcholinesterase activity in various organs. This method is suitable for assessing the toxic effect on the organism. Maybe the researchers could consider usage of different animal models, e.g. chick embryos. This article is clear, understandable for scientists and for readers and I recommends it for publication in present form.

Author Response

Thank you very much for your important observations. We will take into account the acetylcholinesterase activity assay in animal models. We appreciate very much your suggestion. 

Round 2

Reviewer 2 Report

The revised version seems to have improved on the points noted.